# Phosphoproteomics of ATR signaling in mouse testes

**Jennie R Sims**[1†], **Vitor M Faça**[1,2†], **Catalina Pereira**[3†], **Carolline Ascenção**[1], **William Comstock**[1], **Jumana Badar**[1], **Gerardo A Arroyo-Martinez**[3], **Raimundo Freire**[4,5,6], **Paula E Cohen**[3], **Robert S Weiss**[3], **Marcus B Smolka**[1*]

[1]Department of Molecular Biology and Genetics, Weill Institute for Cell and Molecular Biology, Cornell University, Ithaca, United States; [2]Department of Biochemistry and Immunology, Ribeirão Preto Medical School, University of São Paulo, Ribeirão Preto, Brazil; [3]Department of Biomedical Sciences, Cornell University, Ithaca, United States; [4]Unidad de Investigación, Hospital Universitario de Canarias, Tenerife, Spain; [5]Instituto de Tecnologías Biomédicas, Universidad de La Laguna, La Laguna, Spain; [6]Universidad Fernando Pessoa Canarias, Las Palmas de Gran Canaria, Spain

**\*For correspondence:**
mbs266@cornell.edu

[†]These authors contributed equally to this work

**Competing interest:** The authors declare that no competing interests exist.

**Abstract** The phosphatidylinositol 3′ kinase (PI3K)-related kinase ATR is crucial for mammalian meiosis. ATR promotes meiotic progression by coordinating key events in DNA repair, meiotic sex chromosome inactivation (MSCI), and checkpoint-dependent quality control during meiotic prophase I. Despite its central roles in meiosis, the ATR-dependent meiotic signaling network remains largely unknown. Here, we used phosphoproteomics to define ATR signaling events in testes from mice following chemical and genetic ablation of ATR signaling. Quantitative analysis of phosphoproteomes obtained after germ cell-specific genetic ablation of the ATR activating 9-1-1 complex or treatment with ATR inhibitor identified over 14,000 phosphorylation sites from testes samples, of which 401 phosphorylation sites were found to be dependent on both the 9-1-1 complex and ATR. Our analyses identified ATR-dependent phosphorylation events in crucial DNA damage signaling and DNA repair proteins including TOPBP1, SMC3, MDC1, RAD50, and SLX4. Importantly, we identified ATR and RAD1-dependent phosphorylation events in proteins involved in mRNA regulatory processes, including SETX and RANBP3, whose localization to the sex body was lost upon ATR inhibition. In addition to identifying the expected ATR-targeted S/T-Q motif, we identified enrichment of an S/T-P-X-K motif in the set of ATR-dependent events, suggesting that ATR promotes signaling via proline-directed kinase(s) during meiosis. Indeed, we found that ATR signaling is important for the proper localization of CDK2 in spermatocytes. Overall, our analysis establishes a map of ATR signaling in mouse testes and highlights potential meiotic-specific actions of ATR during prophase I progression.

## Editor's evaluation

This paper nicely provides a valuable list of phosphorylation targets by the DNA damage checkpoint kinase, ATR, during mouse spermatogenesis. Importantly, by using the dataset, the authors now showed that ATR controls the localization of CDK2 on meiotic chromosomes, which is critical for meiotic crossover formation.

## Introduction

Meiosis is a specialized cellular process whereby a single round of DNA replication is followed by two successive rounds of cell division to produce haploid gametes. To ensure proper chromosome

segregation at the first meiotic division, meiotic cells must undergo a series of highly regulated processes, including programmed double-strand break (DSB) formation, recombination, and chromosome synapsis (*Keeney et al., 1997*; *Baudat et al., 2013*). The serine/threonine-protein kinase ATR (ataxia telangiectasia and Rad-3 related protein) has well-characterized roles in maintaining genome stability in mitotic cells (*Yazinski and Zou, 2016*; *Saldivar et al., 2017*). In mammals, ATR also plays an essential role in spermatogenesis by promoting meiotic sex chromosome inactivation (MSCI), a process that is required for silencing of the X and Y chromosomes (*Royo et al., 2010*; *Pacheco et al., 2018*; *Widger et al., 2018*). Impairment of ATR activity results in insufficient MSCI and germ cell elimination at the mid-pachytene stage of prophase I (*Widger et al., 2018*; *Royo et al., 2013*; *Turner, 2007*; *Menolfi et al., 2018*; *Fedoriw et al., 2015*). A major readout of ATR activity during MSCI is the phosphorylation of the histone variant H2AX within the dense heterochromatin domain of the nucleus that houses the X and Y chromosomes known as the sex body. Additionally, ATR regulates the sex body localization of several other DNA damage response proteins such as MDC1 and BRCA1, ultimately resulting in MSCI (*Royo et al., 2013*; *Turner et al., 2004*; *Ichijima et al., 2012*). Loss of ATR protein in spermatocytes results in defects in DSB repair and chromosome synapsis, implying that ATR regulates several aspects of meiotic progression (*Menolfi et al., 2018*; *Fedoriw et al., 2015*; *Widger et al., 2018*). Despite the importance of ATR in meiosis, the mechanisms by which meiotic ATR signaling coordinates meiotic progression remain limited due to the complexity and interdependence of meiotic DNA repair, chromosome synapsis, and silencing of unsynapsed chromatin.

ATR activation is relatively well understood in somatic cells, and some of the molecular determinants of ATR activation are shared in meiosis. In mitotic cells, ATR is activated at sites of single-stranded DNA (ssDNA) that arise during replication or DNA repair (*Lovejoy and Cortez, 2009*; *Lempiäinen and Halazonetis, 2009*). ATR is recruited to RPA-coated ssDNA via interaction with ATRIP (ATR interacting protein) (*Zou and Elledge, 2003*), while the 9A-1-1 (RAD9A-RAD1-HUS1) checkpoint clamp is independently loaded at the dsDNA-ssDNA junction by the RAD17-RFC clamp loader (*Delacroix et al., 2007*; *Eichinger and Jentsch, 2011*). The ATR activating protein, TOPBP1 (topoisomerase binding protein 1), then interacts with the c-terminal tail of RAD9A and directly activates ATR via its ATR activation domain (AAD) (*Burtelow et al., 2001*; *Takeishi et al., 2015*; *Roos-Mattjus et al., 2003*; *Thada and Cortez, 2019*; *Kumagai et al., 2006*). Conditional depletion of TOPBP1 in germ cells results in defective MSCI as well as loss of ATR and H2AX phosphorylation in the sex body (*Ellnati et al., 2017*; *Perera et al., 2004*; *Jeon et al., 2019*). The results suggest a prominent role for TOPBP1 in mediating the strong ATR signaling observed during pachynema. Recently, a second ATR activating protein, ETAA1 (Ewing's tumor-associated antigen 1), was identified (*Feng et al., 2016*; *Haahr et al., 2016*; *Lee et al., 2016*; *Bass et al., 2016*). ETAA1 can directly activate ATR at RPA-coated ssDNA at replication forks and is thought to be important for ATR activation during unchallenged DNA replication and unlikely to function in meiosis (*Ellnati et al., 2017*; *Bass and Cortez, 2019*). Once activated, ATR preferentially phosphorylates proteins at S/T-Q motifs (*Kim et al., 1999*; *O'Neill et al., 2000*; *Bastos de Oliveira et al., 2015*). In mitotic cells, ATR-mediated phosphorylation of the histone variant H2AX (hereafter referred to as γH2AX) and the scaffold protein MDC1 contributes to checkpoint activation by enhancing ATR phosphorylation of the downstream kinases CHK1/CHK2, resulting in checkpoint-mediated cell cycle arrest (*Brown and Baltimore, 2003*; *Lou et al., 2003*). In meiosis, ATR also phosphorylates H2AX and MDC1, but, intriguingly, this strong induction of ATR activity observed during normal spermatogenesis is compatible with the progression of the meiotic cell cycle (*Kolas et al., 2005*; *Fernandez-Capetillo et al., 2003*; *Ichijima et al., 2011*). How ATR signaling in meiotic cells coordinates meiotic progression without imposing a checkpoint arrest remains a fundamental unanswered question.

While hundreds of ATR targets have been characterized using quantitative phosphoproteomics mitotic cells (*Bass and Cortez, 2019*; *Lanz et al., 2019*; *Wagner et al., 2016*), much less is understood about ATR signaling in meiosis. A comprehensive dataset of meiotic ATR-dependent phosphorylation events is necessary to further our understanding of the mechanisms by which ATR coordinates DNA repair, chromosome synapsis, checkpoint, and MSCI pathways during meiosis. To define the network of phosphorylation events mediated by ATR in meiotic cells, we performed extensive phosphoproteomic analyses of testes derived from mice with two independent methods of impairing ATR signaling. Given that ATR depletion results in embryonic lethality (*Brown and Baltimore, 2003*; *Brown and Baltimore, 2000*; *de Klein et al., 2000*), we used a genetic model of impairing ATR signaling whereby the 9-1-1

component RAD1 is conditionally depleted under the germ cell-specific *Stra8-Cre*. Depletion of RAD1 in germ cells is anticipated to disrupt all potential 9-1-1 complexes, including those that could form with the testes-specific paralogs Hus1b and *Rad9b* , and therefore significantly impair ATR signaling (*Pereira et al., 2021*). In parallel, we collected testes from mice treated with ATR inhibitor (AZ20) and vehicle-treated controls. We reasoned that by using both a genetic and pharmacological method of inhibiting ATR we would take advantage of both the tissue specificity in the RAD1 conditional knockout as well as the acute effects of rapid ATR inhibition after treatment with ATR inhibitors. By only considering phosphorylation events that are depleted in both the *Rad1* cKO and ATR inhibitor-treated mice, we eliminated pleiotropic effects caused by long-term loss of RAD1, including changes in cellular composition of the testes. After collecting testes from *Rad1* cKO and control mice as well as ATR inhibitor-treated and vehicle control mice, we processed these tissues to isolate phosphorylated peptides, labeled them with amino reactive tandem mass tag reagents (TMT) followed by mass spectrometry to generate phosphoproteomic datasets. By combining these datasets, and after data filtering, we were able to identify a set of 401 phosphorylation events that were dependent on both ATR and RAD1, including phosphorylation of crucial DNA damage signaling and DNA repair proteins at the preferential motif for ATR phosphorylation (S/T-Q). Interestingly, the localization of several of these targets to the sex body, such as Senataxin and RANBP3, was found to be disrupted after ATR inhibition. Finally, the set of ATR- and RAD1-dependent phosphorylation events included enrichment for phosphorylation at the S/T-P-X-K motif, suggesting that ATR promotes signaling of proline-directed kinase(s) in meiosis. Consistent with this model, ATR signaling was important for the proper localization of CDK2 in spermatocytes. Overall, our analysis establishes a comprehensive map of ATR signaling in spermatocytes and highlights potential meiotic-specific actions of ATR during prophase I progression.

## Results

### A combined pharmacological and genetic approach to map ATR-dependent signaling in spermatocytes

Despite the central role of ATR in prophase I of mammalian meiosis, the meiotic ATR signaling network remains largely unknown. Here, we generated phosphoproteomic databases from two sets of mice with independent methods to impair ATR signaling (*Figure 1A and B*). First, to chemically inhibit ATR, we treated mice with the ATR inhibitor AZ20. Next, to genetically impair ATR activity, we utilized a *Rad1* conditional knockout model in which the RAD1 subunit of the 9-1-1 complex is depleted in germ cells under a *Stra8-Cre* promoter. By comparing the datasets generated from these two methods of impairing ATR signaling, we took advantage of the germ cell specificity of the conditional knockout model as well as the rapid dynamics of inhibition with the ATR inhibitor treatment. Our rationale was that phosphorylation impaired both by ATRi treatment and *Rad1* cKO should reflect events that occur early in the ATR response and are specific to germ cells, therefore overcoming limitations intrinsic to each of these experimental setups if analyzed in isolation.

Previous work showed that 7 days of ATRi (AZ20) treatment resulted in strong reduction of γH2AX at the sex body as well as a complete loss of the diplonema population (*Pacheco et al., 2018*). For our phosphoproteomic analyses, to minimize indirect pleiotropic signaling effects and potential changes in testes cellular composition, we asked whether a few hours of AZ20 treatment was enough to inhibit ATR signaling. We collected testes from mice after 4 hr of one dose of AZ20 (50 mg/kg) and examined meiotic chromosome spreads to monitor the localization of γH2AX at the sex body during pachynema, which is dependent on ATR (*Royo et al., 2013*; *Fernandez-Capetillo et al., 2003*; *Ichijima et al., 2011*; *Mahadevaiah et al., 2008*; *Turner et al., 2005*). We observed that 4 hr of ATRi treatment was enough to cause a robust reduction in sex body γH2AX localization in pachynema staged cells (*Figure 1C and D*, *Figure 1—figure supplement 1A and B*). Next, we collected decapsulated testes for mass spectrometry analysis for a total of five pairs of ATRi (4 hr treatment) and vehicle-treated control mice. We also collected decapsulated whole testes from three *Rad1* cKO mice and littermate controls. Tissue samples were lysed and digested with trypsin, followed by enrichment of phosphopeptides and labeling with the 6-plex TMT reagent (*Figure 1B*). HILIC pre-fractionation of TMT-labeled phosphopeptides allowed in-depth quantitative phosphoproteomic analysis, resulting in a total of 49,843 phosphorylation sites identified between the eight experiments (*Figure 1E*). After

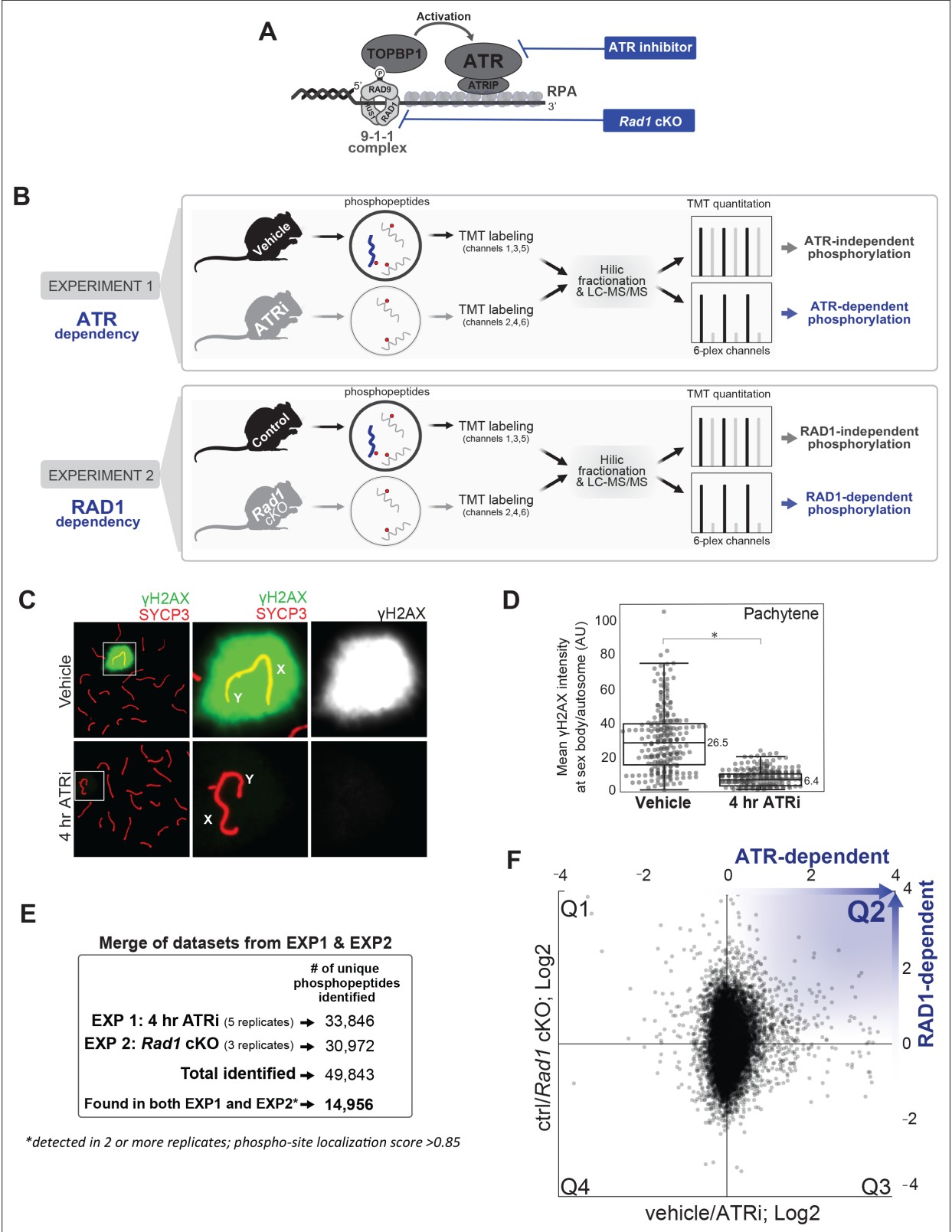

**Figure 1.** Experimental approach for identifying ATR-dependent phosphorylation events in meiosis. (**A**) Schematics depicting the mechanism of ATR activation at a 5′ recessed DNA end via the 9-1-1 complex and TOPBP1, and strategies for chemical and genetic impairment of ATR signaling. (**B**) Whole, decapsulated testes were collected from vehicle and AZ20-treated mice (top) or *Rad1* cKO and control litter mates (bottom) and subjected to quantitative phosphoproteomic analysis to identify ATR-dependent and RAD1-dependent phosphorylation events (see Materials and methods for

*Figure 1 continued*

details). (**C**) Immunofluorescence staining of meiotic spreads from ATRi or vehicle-treated mice collected 4 hr after 50 mg/kg treatment with AZ20. (**D**) Quantification of γH2AX intensity (four vehicle mice, n = 246 cells; four ATRi mice, n = 309 cells p=0.019 measured by Student's *t*-test) (see Materials and methods for more details). (**E**) Description of overall number of replicates and phosphopeptides identified. (**F**) Scatter plot of the final consolidated phosphoproteomic dataset corresponding to the 4 hr ATRi treatment and *Rad1* cKO. Blue in Q2 highlights directionality of ATR and RAD1-dependent phosphorylation events.

The online version of this article includes the following figure supplement(s) for figure 1:

**Figure supplement 1.** Imaging γH2AX on meiotic spreads from vehicle or ATRi-treated mice.

selecting for high-quality phosphosites (localization score >0.85) and considering only phosphopeptides identified in both the AZ20 and *RAD1* cKO datasets in at least two independent replicates, our final list yielded 14,956 quantitated phosphorylation sites (*Figure 1E and F*, *Supplementary file 1*).

## RAD1-dependent ATR signaling targets proteins involved in nucleic acid metabolism, DNA damage response, and the cell cycle

To generate a dataset enriched for early ATR-dependent signaling events that are specific to germ cells, we focused on phosphopeptides displaying consistent reduction in abundance in testes from both ATRi-treated and *Rad1* cKO mice (positioned in quadrant Q2, highlighted in *Figures 1F and 2A*). To exclude data points not consistently altered in both ATRi-treated and *Rad1* cKO datasets, we applied a modified 'bow-tie' filter by excluding data points that fell outside of a fivefold interval of correlation (*Faca et al., 2020*). Next, we applied a filter to remove phosphopeptides with inconsistent quantitative data points to increase confidence of the dataset specifically within the quadrants (see details in Materials and methods). By comparing the number of phosphopeptides present in each of the four quadrants (Q1–Q4), we observed a biased distribution with approximately twofold more differentially phosphorylated peptides in Q2 compared to each of the other quadrants, consistent with a model that the primary mode of ATR activation during meiosis is RAD1-dependent (*Figure 2A and B*, *Figure 2—figure supplement 1A*). Gene enrichment analysis of each filtered quadrant revealed that Q2 is enriched for gene ontology categories such as nucleic acid metabolism, chromosome organization, and cell cycle (*Figure 2C*, *Figure 2—figure supplement 1B*, *Supplementary file 2*), consistent with the expected roles for ATR and RAD1 in these processes. We also assessed the effect of longer ATR inhibition using repeated 50 mg/kg AZ20 treatments over the course of 2.5–3 days and crossed the dataset with the *Rad1* cKO replicates, which yielded 10,881 phosphorylation sites that were common between the ATRi (2.5–3 days) and *Rad1* cKO datasets (*Figure 2—figure supplement 2A*, *Supplementary file 3*). As in the 4 hr treatment dataset, we also observed that Q2 had more sites than in other quadrants, and enrichment for GO categories related to nucleic acid metabolic processes and cell cycle regulation (*Figure 2—figure supplement 2B–D*). Strikingly, Cytoscape/ClueGO analysis revealed that Q2 in both the 4 hr and 2.5–3-day ATRi datasets are enriched in several interconnected categories related to mRNA processing, and that the 2.5–3-day ATRi dataset contains a particularly large group of processes related to cell cycle regulation (*Figure 2—figure supplement 3A–D*).

Depletion of RAD1 in spermatocytes results in reduction in tubule size, infertility, and loss of germ cells (*Pereira et al., 2021*). Meiotic spreads from *Rad1* cKO mice further revealed that spermatocytes have defects in chromosome synapsis and arrest in a pachytene-like stage (*Pereira et al., 2021*). Consistent with these expected effects on downstream events in spermatogenesis, gene enrichment analysis on a subset of RAD1-dependent sites that are not reduced upon 4 hr of ATRi (Q1 in *Figure 2D*, *Figure 2—figure supplement 2E*) revealed enrichment in functional groups related to spermatogenesis, gamete generation, and reproductive processes (*Figure 2E*, *Figure 2—figure supplement 2F*, *Supplementary file 2*). This finding is consistent with our rationale that Q1 would contain phosphorylation events that reflect indirect or pleiotropic events caused by long-term depletion of RAD1, leading to the impairment of processes downstream of meiosis I and loss of cell populations. Gene ontology analysis of RAD1-independent, ATR-dependent phosphosites found in Q3 revealed enrichment of GO terms involved in organelle and cytoskeletal organization (*Figure 2—figure supplement 1B*, *Supplementary file 2*), which could reflect events in non-meiotic cell types.

To further assess the quality of the ATRi and *Rad1* cKO dataset, we examined the distribution of phosphopeptides of MDC1, a known ATR target during prophase I that binds to γH2AX and promotes the spreading of ATR and DNA repair factors to chromatin loops of the X and Y chromosomes to

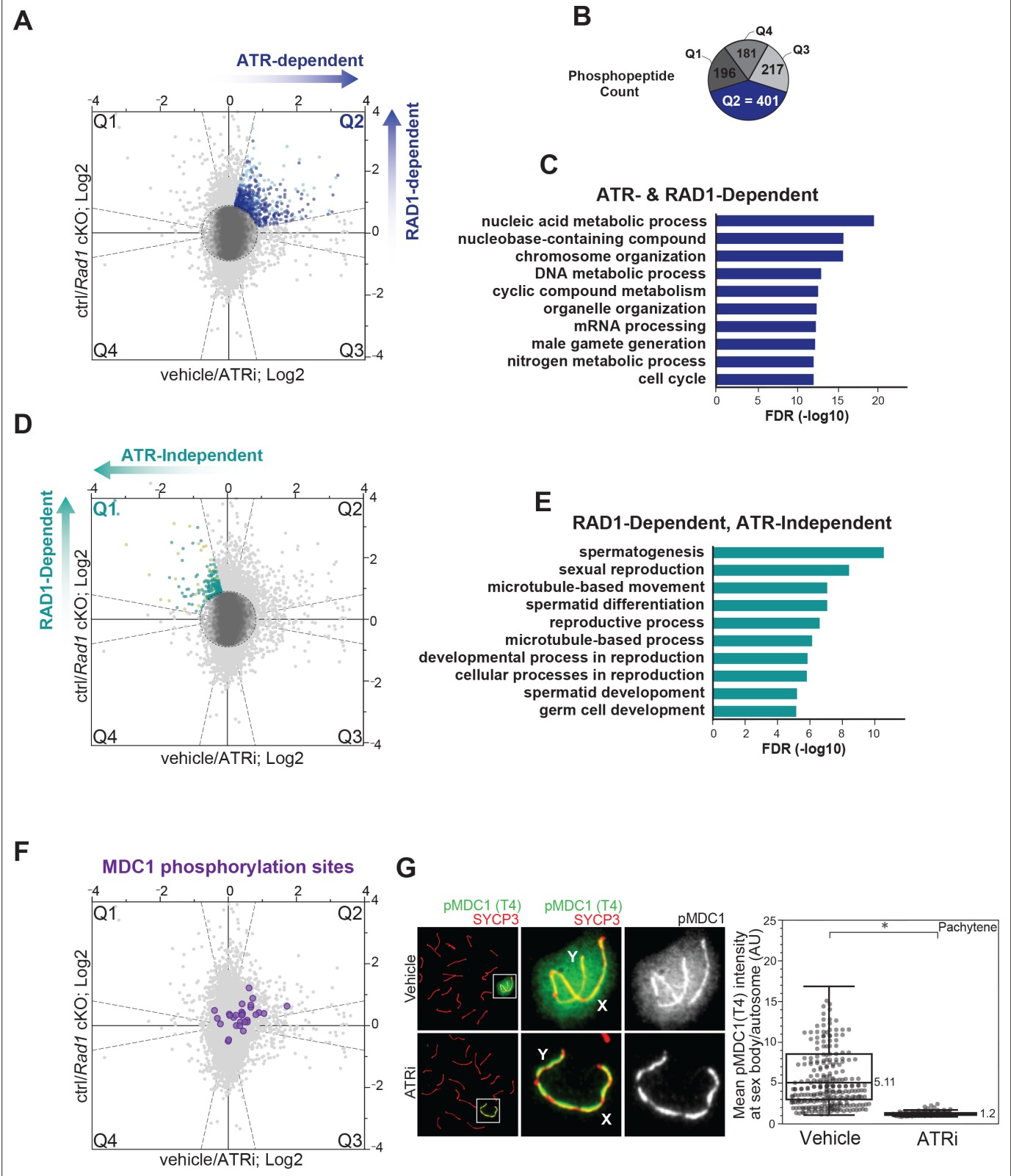

**Figure 2.** ATR and RAD1-dependent signaling events in phosphoproteomic dataset corresponding to 4 hr ATRi treatment and *Rad1* cKO. (**A**) Scatter plot with assignment of phosphopeptides into quadrants delineated by dashed lines ('bow-tie' filter thresholds) and laying outside of a central region ('center' circle) comprised of phosphopeptides considered unchanged in both ATRi and *Rad1* cKO experiments. Quadrant 2 (Q2; blue dots) indicates designated set of ATR and RAD1-dependent phosphopeptides. Phosphorylation sites in each quadrant were also subjected to a filtering step to remove

*Figure 2 continued on next page*

*Figure 2 continued*

inconsistent ratios between replicates of each experiment (see Materials and methods for details). Dark blue dots in Q2 indicate phosphopeptides passing the filter and used for (**B**) and (**C**). Sites that did not pass the filter in Q2 are displayed in light blue. (**B**) Phosphopeptide count per quadrant. (**C**) Curated gene ontology (GO) list of proteins with ATR and RAD1-dependent phosphorylation identified in Q2 from STRING analysis. Full list of GO terms in *Supplementary file 2* Table S3. (**D**) Similar to (**A**), but indicating RAD1-dependent and ATR-independent phosphopeptides in Q1 (pale green) and Q1 sites that passed filtering (teal). (**E**) GO of proteins with RAD1-dependent, ATR-independent phosphorylation from STRING analysis. Full list of GO terms in Table S3 *Supplementary file 2*. (**F**) Scatter plot highlighting the detected MDC1 phosphopeptides. (**G**) Immunofluorescence of meiotic spreads from mice treated with vehicle or 50 mg/kg AZ20 for 4 hr and stained for pMDC1 (MDC1 phosphorylated at threonine 4; green) and SYCP3 (red) with quantification of pachytene-staged cells (four vehicle mice, n = 360 cells; four ATRi mice, n = 282 cells p=0.0268 measured by Student's *t*-test) (see Materials and methods for more details).

The online version of this article includes the following figure supplement(s) for figure 2:

**Figure supplement 1.** Quadrant gene ontology of phosphoproteomic dataset from 4 hr ATRi treatment and *Rad1* cKO.

**Figure supplement 2.** ATR and RAD1-dependent signaling events in phosphoproteomic dataset of 2.5–3-day ATRi treatment and *Rad1* cKO.

**Figure supplement 3.** ClueGO analysis of ATR and RAD1-dependent events in Q2.

**Figure supplement 4.** MDC1 phosphorylation events after ATR inhibition.

promote MSCI (*Ichijima et al., 2011*; *Lou et al., 2006*). We detected 29 MDC1 phosphorylation sites, including 8 sites in Q2 (ATR- and RAD1-dependent) and 21 sites that were not dependent on ATR phosphorylation (*Figure 2F*, *Figure 2—figure supplement 4A*). We also probed the phosphorylation of MDC1 at threonine 4 using a phospho-specific antibody on meiotic spreads and observed a significant loss of signal at the sex body upon 4 hr of ATRi treatment (*Figure 2G*, *Figure 2—figure supplement 4B and C*). These data confirm the effectiveness of the 4 hr ATRi treatment, provide a positive control supporting that Q2 sites contain expected markers of ATR signaling, and reveal that MDC1 undergoes both ATR-dependent and -independent modes of regulation. Furthermore, the identification of MDC1 phosphosites that do not change upon ATR inhibition suggests that the phosphorylation changes in Q2 are unlikely due to changes in MDC1 protein abundance. Notably, we were able to detect at least one phosphorylation site that did not change between the control and reduced ATR activity condition (nonregulated) in over 60% of the proteins containing a phosphorylation site detected in Q2, which indicates that the majority of phosphorylation events in Q2 are not likely due to protein abundance changes, but rather regulated phosphorylation events. Nonetheless, for proteins whose phosphorylation sites are only detected in Q2, it remains possible that protein abundance change may be the underlying cause of the observed change in phosphopeptide abundance. Overall, these results validate our experimental rationale and reinforce the importance of combining data from ATRi and *Rad1* cKO datasets to map meiotic ATR signaling.

## RAD1- and ATR-dependent phosphorylation at S/T-Q sites defines potentially direct ATR targets involved in DNA damage signaling, DNA repair, and RNA metabolism

Given that ATR preferentially phosphorylates S/T-Q motifs (*Kim et al., 1999*; *O'Neill et al., 2000*), we reasoned that most phosphorylation sites at S/T-Q in Q2 are more likely to reflect direct ATR substrates. Notably, the set of Q2 sites with an S/T-Q motif was enriched in proteins involved in DNA repair, including MDC1, UIMC1 (RAP80), and the components of the MRN complex RAD50 and NBN (NBS1) (*Figure 3A–C*). The group of S/T-Q sites in Q2 also included proteins involved in RNA metabolism and chromatin regulation (*Figure 3D and E*, *Figure 3—figure supplement 1A and B*). Notably, S/T-Q sites in proteins involved in RNA metabolism (SETX, XPO5, and RANBP3) displayed high ATR dependency (*Figure 3D*, *Figure 3—figure supplement 1C and D*) and were detected in both the 4 hr and 2.5–3-day ATRi datasets (*Figure 3—figure supplement 2A and B*, *Supplementary file 4*). Most proteins with an S/T-Q phosphorylation site identified in Q2 have additional phosphorylation sites that did not change in the *Rad1* cKO or ATR inhibition (*Figure 3E*, *Figure 3—figure supplement 1A*), indicating that the overall abundance of these proteins was not likely changing. Furthermore, many Q2 proteins contained both an S/T-Q motif and another non-S/T-Q phosphorylation site that was also ATR-dependent (*Figure 3E*, *Figure 3—figure supplement 1A*). Taken together, these results reveal a set of proteins that may potentially be direct substrates of ATR in meiosis and define a set of RNA regulatory proteins subjected to 9-1-1 and ATR-dependent phosphorylation.

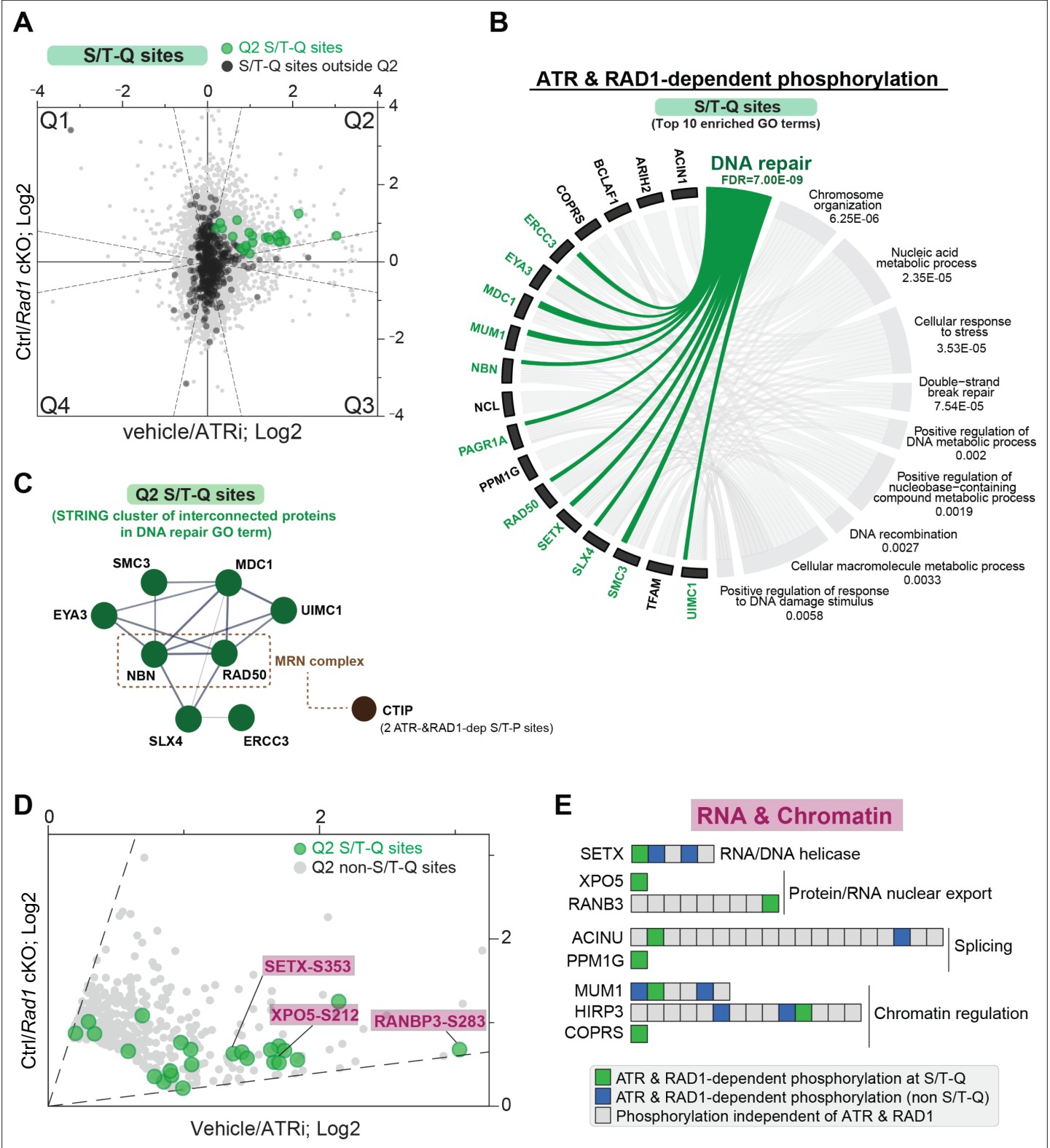

**Figure 3.** RAD1 and ATR-dependent phosphorylation at the S/T-Q motif. (**A**) Scatter plot highlighting all S/T-Q phosphorylation outside Q2 (dark gray) and S/T-Q phosphorylation inside Q2 (green). (**B**) Chord diagram of gene ontology of ATR and RAD1-dependent S/T-Q phosphorylation events was done using STRING-db network functional enrichment analysis. The top 10 significantly enriched biological processes GO terms were selected and represented as a chord diagram. GO terms are shown on the right and proteins found for each term on the left. False discovery rate (FDR) for GO term enrichment is shown below each term. (**C**) STRING network of interconnected DNA repair proteins with Q2 S/T-Q phosphorylation. Non-S/T-Q sites for

*Figure 3 continued on next page*

*Figure 3 continued*

the MRN-related protein CTIP were also present in Q2, suggesting that it is under regulation by a proline-directed kinase controlled by ATR. (**D**) Scatter plot of data shown in (**A**) highlighting Q2 S/T-Q phosphopeptides in proteins involved in RNA metabolism. (**E**) Selected set of proteins involved in chromatin modification and RNA metabolic processes with all identified phosphorylation sites ordered sequentially from the n-terminus to the c-terminus of each protein.

The online version of this article includes the following figure supplement(s) for figure 3:

**Figure supplement 1.** RAD1 and ATR-dependent phosphorylation at the S/T-Q motif in the 2.5–3-day ATRi vs. *Rad1* cKO dataset.

**Figure supplement 2.** Comparison of 2.5–3-day ATRi and 4 hr ATRi treatment Q2 datasets.

## ATR modulates the localization of RNA regulatory factors Senataxin and RANBP3

Although ATR localizes to the sex body to promote MSCI, it is not known if ATR directly regulates RNA metabolic proteins to promote silencing or processing of RNAs. To investigate how meiotic ATR may regulate RNA metabolism, we focused on RNA metabolic proteins with S/T-Q phosphosites in Q2. We found serine 353 in Senataxin (SETX), an RNA:DNA helicase with established roles in transcriptional regulation and genome maintenance (*Cohen et al., 2018*; *Groh et al., 2017*) to be downregulated upon RAD1 loss and ATR inhibition (*Figure 3D*, *Figure 3—figure supplement 1D*). Senataxin disruption is associated with male infertility in humans, and *Setx⁻/⁻* male mice are infertile resulting from arrest at the pachytene stage in meiosis I (*Becherel et al., 2019*; *Becherel et al., 2013*). Senataxin localizes to the XY chromosomes and promotes the localization of ATR, γH2AX, and other DNA repair and checkpoint factors to the sex body during MSCI (*Yeo et al., 2015*). Senataxin interacts with many proteins involved in transcription and is thought to regulate multiple aspects of RNA metabolism such as splicing efficiency and transcription termination in part by its activity in resolving R-loops (RNA-DNA hybrids) (*Skourti-Stathaki et al., 2011*). To assess whether ATR modulates Senataxin function in meiosis, we stained for Senataxin in meiotic spreads derived from both ATR inhibitor-treated and *RAD1* cKO mice. In accordance with previous work, we found that Senataxin localizes to the sex body at pachynema control spreads (*Becherel et al., 2013*; *Yeo et al., 2015*). Strikingly, Senataxin accumulation at the sex body was significantly reduced in pachytene spreads derived from ATRi-treated mice (*Figure 4A and B*, *Figure 4—figure supplement 1A and B*). While it is difficult to morphologically distinguish the X and Y chromosomes in the *Rad1* cKO spreads, we observed no enrichment of Senataxin around any selection of chromosomes in pachytene-like spreads with four or more synapsed autosomes in both 8–12-week and 14-day-old mice (*Figure 4C and D*, *Figure 4—figure supplement 1C and D*). Previous studies have found that Senataxin inhibition results in diminished ATR signaling at the sex body and suggested a role for Senataxin in promoting ATR signaling (*Becherel et al., 2013*; *Yeo et al., 2015*). Our results further suggest that ATR promotes the recruitment or retention of Senataxin at the sex body, consistent with a model in which Senataxin and ATR act in a feed-forward loop to cooperatively promote their recruitment and efficient sex body formation and MSCI.

Another protein with a S/T-Q phosphorylation site downregulated upon RAD1 loss and ATR inhibition was RANBP3 (phosphorylation at serine 283) (*Figure 3D and E*, *Figure 3—figure supplement 1D*). RANBP3 is a relatively unknown protein with connections to RNA and protein nucleo-transport (*Boudhraa et al., 2020*). Unfortunately, it is not known if RANBP3 depletion results in a loss of fertility although one study has found an association between decreased RANBP3 expression and human infertility (*Tang et al., 2020*). We investigated the localization of RANBP3 in meiotic spreads and found that in cells derived from wild-type or vehicle-treated mice, RANBP3 localizes to the sex body at pachynema (*Figure 4E and F*, *Figure 4—figure supplement 2A and B*). The accumulation of RANBP3 is significantly lost at the sex body derived from ATRi-treated mice and at all chromosome cores in *Rad1* cKO pachytene-like spreads in both 8–12-week and 14-day-old mice (*Figure 4G and H*, *Figure 4—figure supplement 2C and D*), suggesting a role for ATR in the recruitment or retention of RANBP3 at the sex body. Overall, these results support a model whereby ATR promotes the proper localization of SETX and RANBP3 to the sex body in pachynema.

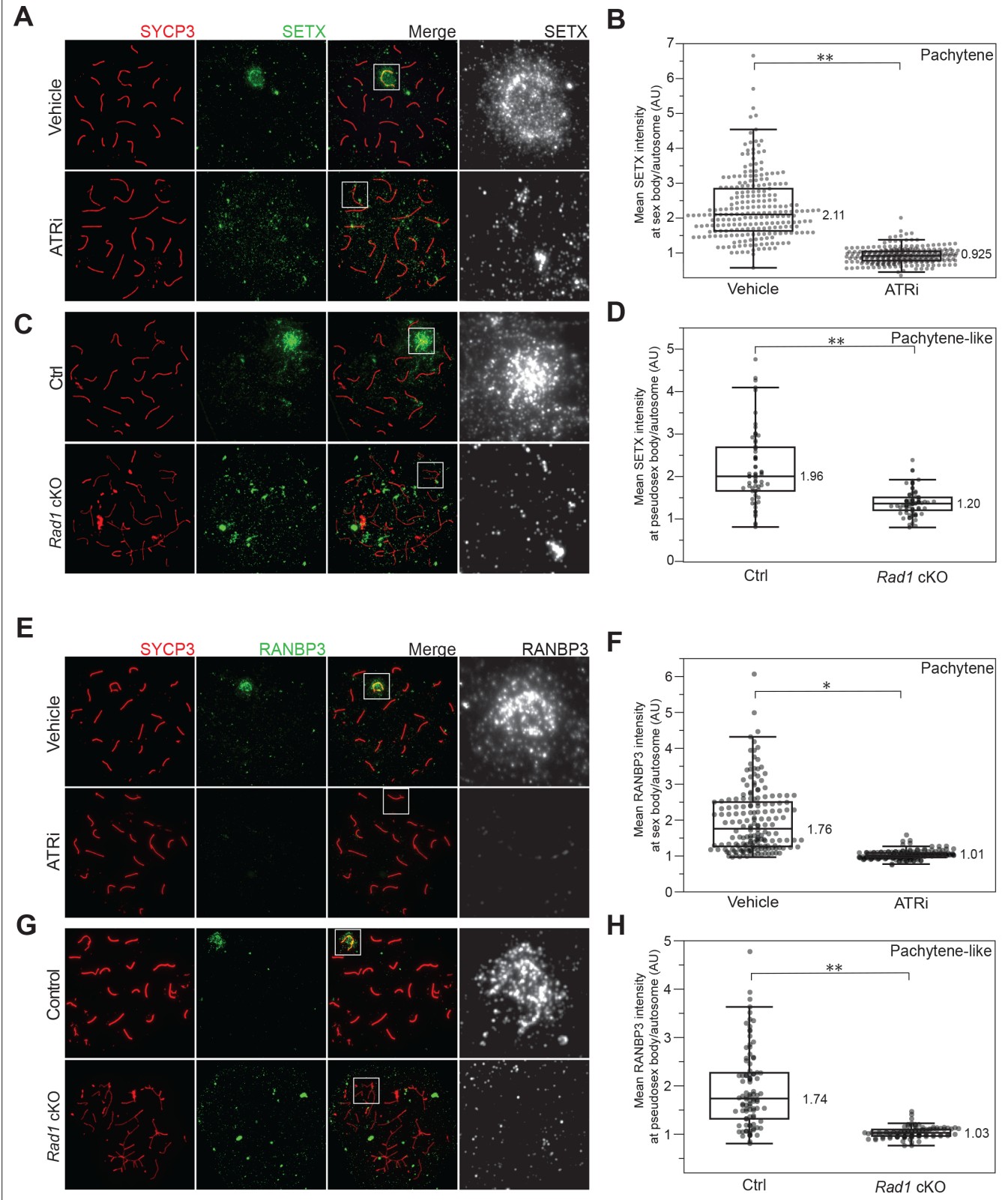

**Figure 4.** Senataxin (SETX) and RANBP3 localization in meiotic spreads after ATR inhibition. (**A**) Immunofluorescence of meiotic chromosome spreads with SETX (green) and SYCP3 (red) from mice collected 4 hr after 50 mg/kg treatment with AZ20 or vehicle. (**B**) Quantification of pachytene spreads in a (four vehicle mice; n = 237 cells; four ATRi mice; n = 283 cells p=0.00435 measured by Student's *t*-test). (**C**) *Rad1* cKO and control spreads stained as in (**A**). (**D**) Quantification of pachytene or pachytene-like spreads in (**C**) (four control mice, n = 64 cells; four RAD1 cKO mice, n = 72 cells p=0.00286

*Figure 4 continued on next page*

Figure 4 continued

measured by Student's *t*-test). (**E**) Immunofluorescence of meiotic chromosome spreads with RANBP3 (green) and SYCP3 (red) from mice collected 4 hr after 50 mg/kg treatment with AZ20 or vehicle. (**F**) Quantification of pachytene spreads in e (three vehicle mice, n = 174 cells; three ATRi mice, n = 167 cells p=0.048 measured by Student's *t*-test). (**G**) *Rad1* cKO and control spreads stained as in (**E**). (**H**) Quantification of pachytene or pachytene-like spreads in (**G**) (four control mice, n = 96 cells; four cKO mice, n = 99 cells p=0.0039 measured by Student's *t*-test).

The online version of this article includes the following figure supplement(s) for figure 4:

**Figure supplement 1.** Effect of ATR inhibition on Senataxin (SETX) localization in meiotic spreads.

**Figure supplement 2.** Effect of ATR inhibition on RANBP3 localization in meiotic spreads.

## Meiotic ATR promotes extensive phospho-signaling at an S/T-P-X-K motif

Despite phosphorylation at the S/T-Q motif being enriched in Q2, it still represented only a small portion of the RAD1- and ATR-dependent sites identified in both datasets for 4 hr and 2.5–3 days of ATRi (*Figure 5A*, *Supplementary files 1 and 3*). The finding that most phosphorylation sites in Q2 are not in the S/T-Q motif suggests that ATR is able to phosphorylate other motifs and/or directly or indirectly regulate the activity of other kinases or phosphatases during meiosis. To better characterize the set of non-S/T-Q phosphorylation events that are dependent on ATR and RAD1, we searched for other motifs enriched in Q2 in the datasets of both 4 hr and 2.5–3 days of ATR inhibition. First, analysis of the amino acid at the +1 position following the phosphorylated S/T residue revealed that a large fraction of phosphorylation sites contain a proline at the +1 position (*Figure 5A*). Interestingly, the proportion of S/T-P sites in Q2 increased from ~42% in the 4 hr ATRi dataset to over 65% in the 2.5–3-day ATRi dataset, suggesting that these sites could represent the impairment of downstream signaling events mediated indirectly by a kinase or phosphatase regulated by ATR, and would therefore take longer to be impaired after ATR inhibition. Since S/T-P is a rather common motif in the phosphoproteome, we searched for a more specific motif that could reveal the potential identify of the kinase or phosphatase involved. We computed the relative proportion of each amino acid at the ±6 positions surrounding the identified phosphorylation sites, comparing their prevalence in Q2 (ATR and RAD1-dependent sites) versus center (unregulated or not-differentially phosphorylated sites) (*Figure 5—figure supplement 1A and B*). The matrices for both the 4 hr and 2.5–3-day ATRi treatment datasets revealed enrichment for K at +3 position in the set of ATR and RAD1-dependent sites. Close inspection of the group of ATR and RAD1-dependent phosphosites with K at +3 revealed that most contain a P at the +1 position, both for the 4 hr and 2.5–3-day ATRi datasets (*Figure 5—figure supplement 1C*), suggesting the enrichment of an S/T-P-X-K motif. Indeed, comparison of the prevalence of S/T-P-X-K motifs in Q2 (ATR- and RAD1-dependent) to the set of unregulated phosphosites found in the center of the dataset revealed substantial enrichments of the S/T-P-X-K motif in both the 4 hr and 2.5–3-day ATRi Q2 datasets (*Figure 5B*). The enrichment was greater in the 2.5–3-day ATRi dataset, reaching over 15% of all Q2 sites, as compared to comprising only 3% in the center (unregulated) sites. These results suggest that ATR is regulating the activity of one or more kinases that have a preference for S/T-P-X-K motif. Notably, this motif fits the canonical preferential motif for the master cell cycle kinases CDK1 and CDK2. We were able to obtain an antibody against CDK2 that allowed us to visualize the localization of this kinase in meiotic chromosome spreads. We observed that CDK2 localizes to the core as well as to the ends of chromosomes, similar to previously reported (*Tu et al., 2017*). Image analysis revealed that autosomal localization of CDK2 is significantly disrupted by ATRi, and surprisingly, this effect could be observed already at 4hr after ATRi treatment (*Figure 5C and D*, *Figure 5—figure supplement 2A and B*). Importantly, 4 hr of ATRi treatment does not result in major changes in cellular composition (*Figure 5—figure supplement 3A and B*), supporting that the changes in S/T-P-X-K sites observed at 4 hr after ATRi are likely due to specific alterations in an ATR signaling axis rather than indirect effects due to gross changes in testis cellularity. These results are consistent with the finding that the several phosphorylation events at the canonical S/T-P-X-K motif were depleted after ATRi. The identification of more Q2 S/T-P-X-K sites in the 2.5–3-day ATRi dataset when compared to the 4 hr ATRi dataset suggests that different downstream S/T-P-X-K are likely under distinct dynamics of phosphorylation and dephosphorylation following ATRi. CDK2 has established roles in cell cycle regulation and, in the context of meiosis, is critical for meiotic prophase I progression (*Berthet et al., 2003*; *Ortega et al., 2003*; *Chauhan et al., 2016*; *Palmer et al., 2020*; *Singh et al., 2019*). Consistent with

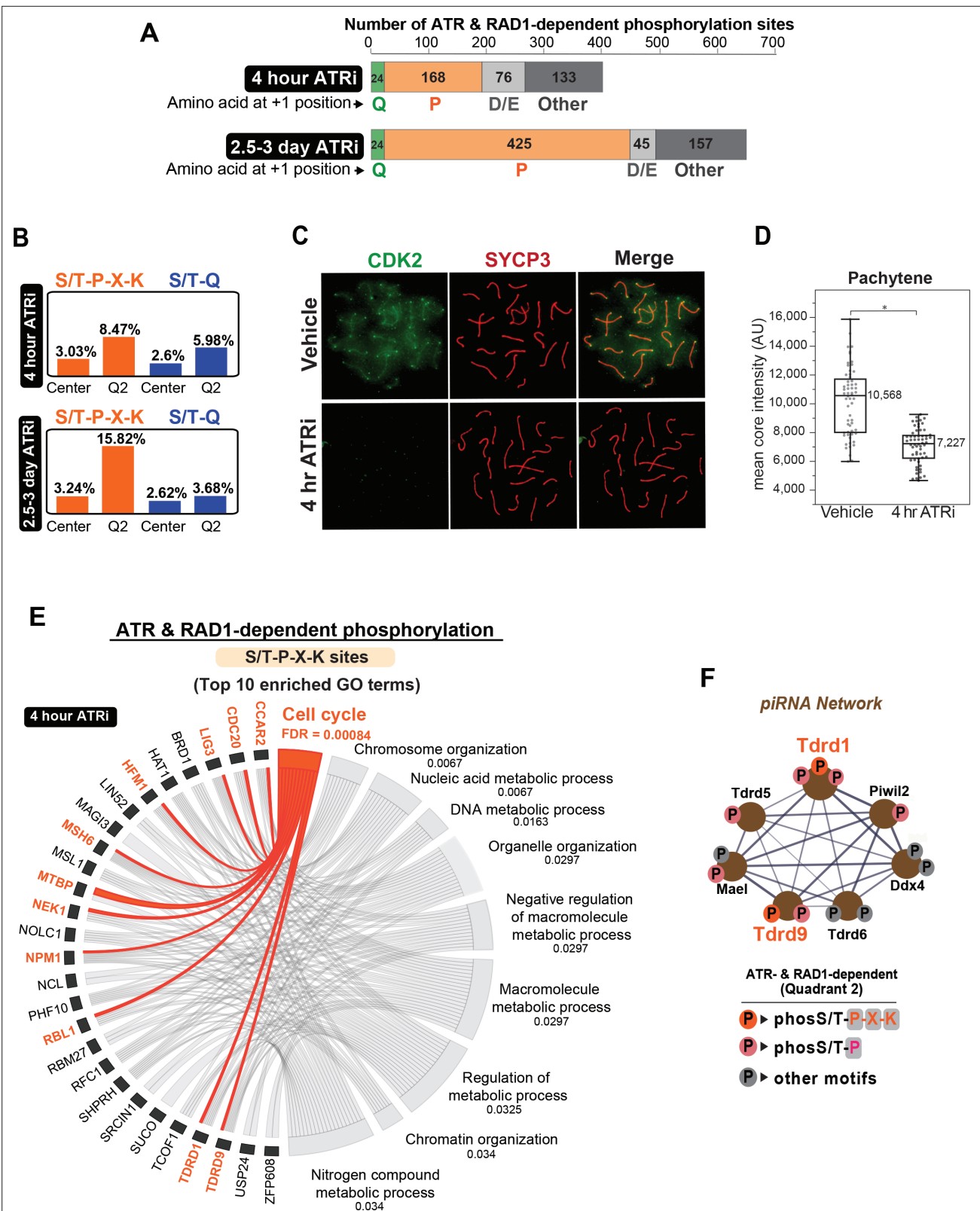

**Figure 5.** Enrichment of S/T-P-X-K phosphorylation motif in the set of ATR and RAD1-dependent signaling events. (**A**) Bar graph depicting the count of Q2 phosphopeptides with the indicated amino acids at the +1 position. (**B**) Bar graph of the percentage of indicated phospho-motifs in the center (unchanged events) and Q2. (**C**) Immunofluorescence of meiotic spreads from mice treated with vehicle or 50 mg/kg AZ20 for 4 hr and stained for CDK2 (green) and SYCP3 (red). (**D**) Quantification of CDK2 signal intensity of pachytene-staged cells in (**C**) (three vehicle mice, n = 60 cells; three ATRi mice,

*Figure 5 continued on next page*

*Figure 5 continued*

n = 60 cells p=0.0368 measured by Student's *t*-test) (see Materials and methods for more details). (**E**) Chord plot of gene ontology (GO) analysis of ATR and RAD1-dependent S/T-P-X-K phosphorylation events using STRING. The top 10 significantly enriched biological processes GO terms were selected. False discovery rate (FDR) for GO term enrichment is shown below each term. GO terms are shown on the right and proteins found for each term on the left. (**F**) STRING analysis cluster of piRNA-related proteins with Q2 phosphorylation.

The online version of this article includes the following figure supplement(s) for figure 5:

**Figure supplement 1.** RAD1- and ATR-dependent signaling is enriched for phosphorylation events at the S/T-X-X-K motif.

**Figure supplement 2.** Effect of ATRi treatment on the localization of CDK2 in meiotic spreads.

**Figure supplement 3.** Histological and PCA analysis of ATRi treated mice.

the model that CDK2 activity is impaired by ATRi and RAD1 loss of function, gene ontology analysis of proteins with S/T-P-X-K motif phosphorylation in quadrant 2 revealed a significant enrichment for genes involved in regulation of the cell cycle, including CDC20 and RBL1 (*Figure 5E*). A range of S/T-P sites without K at +3 are also present in Q2 and could still reflect other potential substrates of CDK2, CDK1, or other proline-directed kinases regulated by ATR (*Supplementary files 1 and 3*). While the loss of proper CDK2 localization at pachynema in ATRi-treated mice supports a potential mechanistic link between ATR and CDK2, further work will be needed to understand the role of meiotic ATR signaling in regulating CDK2.

Interestingly, we noticed that several proteins involved in the biogenesis of piRNAs were phosphorylated at S/T-P sites in a RAD1- and ATR-dependent manner (*Figure 5F*). piRNAs are known to play key roles in spermatogenesis by preventing retrotransposon integration during meiosis (*Goh et al., 2015*; *Marcon et al., 2008*), and male mice deficient for genes involved in piRNA biogenesis are infertile due to spermatocyte arrest (*Fu and Wang, 2014*). We detected depletion of S/T-P phosphorylation sites (including two S/T-P-X-K sites) in the piRNA factors TDRD1, TDRD9, TDRD5, MAEL, and PIWIL2 (*Figure 5F*). Additional piRNA-related proteins contained ATR- and RAD1-dependent phosphorylation at non-S/T-P motifs, including DDX4, TDRD6, and MAEL. While these data suggest the action of ATR-regulated kinases on the regulation of piRNA-related proteins, we do not exclude that the observed depletion of some of the phosphorylation sites may be due, in part, to changes in protein abundance. Further work will be important to better understand how ATR controls the network of piRNA proteins.

## Discussion

ATR has well-established roles in promoting genome stability in mitotic cells by regulating multiple aspects of DNA metabolism such as DNA repair, DNA replication, and the DNA damage checkpoint (*Lanz et al., 2019*; *Pereira et al., 2020*). Several phosphoproteomic databases have been generated to characterize the targets of ATR during conditions of replication stress or within the context of mitosis (*Bass and Cortez, 2019*; *Bastos de Oliveira et al., 2015*; *Wagner et al., 2016*; *Lanz et al., 2018*; *Schlam-Babayov et al., 2021*; *Matsuoka et al., 2007*). These resources have been useful not only to mechanistically dissect the different roles of ATR, but also to gain a more comprehensive understanding of its multifaceted action in genome metabolism. In the context of meiosis, much less is understood about ATR signaling, and although previous reports have catalogued phosphorylation events in mouse testis using phosphoproteomics, these datasets lack experimentally established kinase-substrate relationships (*Guo et al., 2008*; *MacLeod et al., 2014*; *Castillo et al., 2019*; *Qi et al., 2014*; *Li et al., 2019*). Given the utmost importance of defining the ATR-mediated signaling events in mammalian meiosis for allowing mechanistic dissection of its function and mode of action, here we performed an in-depth phosphoproteomic analysis of ATR signaling in testes. The success of our work mostly relied on a two-part approach for identifying high-confidence ATR-dependent phosphorylation events. By combining the datasets from the *Rad1* cKO genetic mouse model and ATR inhibitor-treated mice, we enhanced our confidence in identifying meiosis-enriched ATR-dependent signaling events. Importantly, this dual approach allowed us to remove phosphorylation events changing as a result of loss of spermatocyte and sperm populations in *Rad1* cKO testes. Importantly, we did not observe gross changes in cellular composition after 4 hr of ATRi treatment (*Figure 5—figure supplement 3A and B*), supporting that the observed changes in phosphorylation events correspond to specific impacts on

ATR-mediated signaling axis rather than pleiotropic changes in testis cellularity or persistent cell cycle arrest. Furthermore, the choice of AZ20 as the ATR inhibitor in this study allowed us to specifically study the effect of loss of ATR signaling without altering ATM or DNA-PKcs signaling, as AZ20 is a highly selective inhibitor of ATR (*Foote et al., 2013*). As validation of our dataset, we detected known ATR targets such as MDC1 in the set of ATR and RAD1-dependent signaling events. Additionally, we observed the expected enrichment for DNA metabolism, DNA repair, and cell cycle gene ontology categories. While we did not detect phosphorylation events in some established targets of ATR such as CHK1, this is likely a reflection of the limitations of generating complete coverage from whole mouse testis and the low abundance of some checkpoint proteins. Another limitation is the intrinsic variability of the dataset, mainly in the datasets of ATRi-treated testes. While the *Rad1* cKO datasets clustered tightly together in a principal component analysis (PCA), the ATRi conditions showed more variability along principal component 2, which could reflect the difficulties in reproducibly delivering ATR inhibitors to the testis via oral/systemic administration (*Figure 5—figure supplement 3C*). Despite these noted limitations, we anticipate that the dataset presented here will serve as a framework for the meiosis community to mechanistically dissect distinct actions of meiotic ATR.

Our findings revealed interesting new connections between ATR and RNA metabolism, as illustrated by the detection of ATR- and RAD-dependent phosphorylation in Senataxin and RANBP3. It is tempting to speculate that these are direct functional targets since they were phosphorylated at the S/T-Q motif and their localization to the XY body was compromised during ATR inhibition. Since RNA metabolism and ATR signaling are both closely linked to the establishment of transcriptional silencing at the XY, our data suggest that ATR-mediated phosphorylation of Senataxin, RANBP3, and other RNA processing factors may play central roles in promoting proper silencing and proper meiotic progression (*Figure 5—figure supplement 3D*). The connection of ATR to RNA metabolism is not completely surprising since it was previously reported in the context of mitotic cells, although the targets and mechanisms remain poorly understood (*Burger et al., 2019*). Notably, given that silencing of the XY is inextricably linked to prophase I progression, it is likely that the connection of meiotic ATR signaling to RNA metabolism is even more relevant compared to its mitotic signaling. An interesting model to be explored in future work is that SETX and RANBP3 may coordinate the removal of RNA from XY DNA to establish MSCI. Further work using genetic models for phosphorylation site mutations will be needed to establish the specific role of ATR-mediated phosphorylation of these proteins and to dissect the mechanism by which ATR promotes the localization and action of Senataxin, RANBP3, and potentially other RNA metabolic proteins identified in this study.

Since our phosphoproteomic is unbiased and not only directed at the preferred S/T-Q motif, we were able to capture a range of phosphorylation events in other motifs suggesting that ATR regulates multiple downstream kinases during meiosis. Strikingly, we observed a strong enrichment for ATR-dependent phosphorylation sites at the S/T-P-X-K motif (a preferred consensus motif for CDK1 and CDK2) and found that ATR signaling is important for proper localization of CDK2 to autosomes. A simple model would predict that ATR somehow activates CDK2 during prophase I (*Figure 5—figure supplement 3D*), which could be tested by future phosphoproteomic analysis of testes from mice treated with CDK2 inhibitors. We were unable to test the effect of ATR inhibition on the localization of CDK1 due to lack of a proper antibody; however, we do not exclude that similar to CDK2 the localization and action of CDK1 may also be affected by ATR inhibition. It is worth mentioning that in mitosis the canonical action of ATR in promoting DNA damage checkpoint, and consequent cell cycle arrest, is mediated via inhibition CDK activity, and consequent reduction in S/T-P phosphorylation sites (*Sørensen and Syljuåsen, 2012*). In this sense, the observed dependency of S/T-P-X-K motif for ATR in meiosis is the opposite to what would be predicted from mitotic cells. Since the high activity of ATR in meiosis does not result in meiotic arrest, but is actually required for meiotic progression, it is possible that our data is revealing a drastic difference in how ATR signaling is wired with downstream kinases in meiotic versus mitotic cells. In addition to S/T-P-X-K motif, several other motifs were represented in the set of ATR- and RAD1-dependent sites. We cannot exclude that the effect of ATR inhibition in depleting a range of phosphorylation events could be due, at least in part, to a function of ATR in regulating phosphatases. Importantly, phosphorylation of phosphatases, including PPM1G and PP1R7, was found to be reduced upon ATR inhibition and *Rad1* cKO. A potential scenario is that ATR inhibits constitutively active phosphatases, so ATR inhibition would lead to increased phosphatase activity and reduced phosphorylation of their targets. Interestingly, the fact that phosphorylation

of H2AX in the sex body is drastically reduced already 4 hr after ATR inhibition strongly suggests the existence of a constant phosphorylation cycle involving ATR kinase activity and the constitutive action of phosphatases. Further investigations of this phospho-cycle could lead to important new insights about the dynamics and roles of phosphorylation circuitries during prophase I.

Overall, our work represents an initial attempt to reveal the scope of targets and processes affected by meiotic ATR signaling. As expected, the ATR signaling network in meiosis is overwhelmingly complex and multifaceted. Major challenges remain, especially to untangle the functional relevance of most of the identified signaling events, and understand how the different modes of ATR signaling are coordinated for proper control of meiotic progression. Another key outstanding question is to understand how the ATR kinase, which imposes cell cycle checkpoints in most other cell types, is so highly activated in spermatocytes without inducing cell cycle arrest. A potential explanation may lay at the specificity of ATR's action at the sex body, which may be devoted to the regulation of checkpoint-independent processes such as the control of RNA processing during meiotic prophase I (*Figure 5—figure supplement 3D*), as supported by our data. Finally, there are medical implications of understanding ATR signaling in meiosis since many ATR inhibitors are currently in phase 2 clinical trials for cancer treatment and determining the impact of these inhibitors in meiotic cells will be relevant to define the effects of these treatments in patient fertility.

# Materials and methods

## ATR inhibitor treatment of mice

AZ20 was reconstituted in 10% DMSO (Sigma), 40% propylene glycol (Sigma), and 50% water. Control mice were treated with 10% DMSO (Sigma), 40% propylene glycol (Sigma), and 50% water. Wild-type C57BL/6 male mice aged to 8–12 weeks old were gavaged with 50 mg/kg of AZ20 (Selleckchem) and euthanized at indicated time points. Specific time points examined in this study include collection after 2.5 or 3 days of 50 mg/kg of AZ20 per day or 4 hr after one dose of 50 mg/kg of AZ20.

## Enrichment of testes phosphopeptides and TMT labeling

Whole, decapsulated testes were collected and frozen at –80°C from 8- to 12-week-old AZ20 and vehicle-treated C57BL/6 mice 4 hr after treatment as indicated. To generate the *Rad1* cKO mice used in this study, *Rad1^+/-^, Stra8-Cre^+^* mice were crossed with *Rad1^flox/flox^* mice to generate *Rad1^-/fl^*, *Stra8-Cre^+^* animals. Whole, decapsulated testes from these *Rad1* cKO and littermate controls on a 129Sv/Ev background were collected at 8 or 12 weeks (*Pereira et al., 2021*). *Rad1* cKO control genotypes are *Rad1^-/fl^,Stra8-Cre^-^; Rad1^+/fl^; Stra8-Cre^-^* and *Rad1^+/fl^;Stra8-Cre^+^*. Individual testes were thawed at 4°C in lysis buffer (50 mM Tris pH 8.0, 5 mM EDTA, 150 mM NaCl, 0.2% Tergitol) supplemented with 1 mM PMSF and PhosSTOP (sigma) and sonicated. 4 mg of protein (quantified by Bradford protein assay, Bio-Rad) was collected, denatured with 1% SDS, and reduced with 5 mM DTT at 65°C for 10 min followed by alkylation with 60 mM iodoacetamide. Proteins were precipitated in a cold solution of 50% acetone, 49.9% ethanol, and 0.1% acetic acid, and protein pellet was washed once with a solution of 0.08 M urea in water. Protein pellet was resuspended at a concentration of 10 mg/ml in 8 M urea, Tris 0.05 M, pH 8.0, NaCl 0.15 M, and diluted fivefold prior digestion with 80 µg of trypsin (TPCK-treated, Sigma) overnight at 37°C. Protein digests were acidified to a final concentration of 1% TFA and cleaned up in a solid-phase extraction (SPE) C$_{18}$ cartridge pre-conditioned with 0.1% TFA solution. Peptides were eluted from the cartridges with 80% acetonitrile, 0.1% acetic acid aqueous solution, and dried in a speedvac. Phosphopeptide enrichment was performed using a High-Select Fe-NTA Phosphopeptide Enrichment Kit according to the manufacturer's protocol (Cat# A32992, Thermo Scientific). Phosphopeptide samples were split into four aliquots (10, 30, 30, and 30%) and dried in salinized glass insert tubes. For each experiment, the three 30% aliquots from each control and AZ20-treated, or control and *Rad1* cKO, samples were resuspended in 35 µL of 50 mM HEPES and labeled with 100 µg of each of the TMT sixplex Isobaric Label Reagents (Thermo Scientific), previously diluted in 15 µL of pure acetonitrile. TMT-labeling reaction was carried out at room temperature for 1 hr and quenched with 50 µL of 1 M glycine. All sixplex TMT-labeled aliquots were mixed, diluted with 200 µL of aqueous solution of formic acid 1% (v/v), and cleaned up in a SPE 1 cc C$_{18}$ cartridge (Sep-Pak C18 cc vac cartridge, 50 mg Sorbent, WAT054955, Waters). Bound TMT-labeled phosphopeptides were eluted with 50% acetonitrile, 0.1% formic acid in water, and dried in a speedvac.

## Mass spectrometric analysis of TMT-labeled phosphopeptides

Dried TMT-labeled phosphopeptides were resuspended in 16.5 µL water, 10 µL formic acid 10% (v.v), and 60 µL of pure acetonitrile and submitted to HILIC fractionation prior to mass spectrometry analysis in a TSK gel Amide-80 column (2 mm × 150 mm, 5 µm; Tosoh Bioscience) using a three-solvent system: buffer A (90% acetonitrile), buffer B (75% acetonitrile and 0.005% trifluoroacetic acid), and buffer C (0.025% trifluoroacetic acid). The chromatographic runs were carried out at 150 µL/min and gradient used was 100% buffer A at time = 0 min; 94% buffer B and 6% buffer C at t = 3 min; 65.6% buffer B and 34.4% buffer C at t = 30 min with a curve factor of 7; 5% buffer B and 95% buffer C at t = 32 min; isocratic hold until t = 37 min; 100% buffer A at t = 39–51 min. 1 min fractions were collected between minutes 8 and 10 of the gradient; 30 s fractions between minutes 10 and 26; and 2 min fractions between minutes 26 and 38 for a total of 40 fractions. Individual fractions were combined according to chromatographic features, dried in a speedvac, and individually submitted to LC-MS/MS analysis. Individual phosphopeptide fractions were resuspended in 0.1% trifluoroacetic acid and subjected to LC-MS/MS analysis in an UltiMate 3000 RSLC nano chromatographic system coupled to a Q-Exactive HF mass spectrometer (Thermo Fisher Scientific). The chromatographic separation was carried out in 35-cm-long 100 µm inner diameter column packed in-house with 3 µm $C_{18}$ reversed-phase resin (Reprosil Pur C18AQ 3 µm). Q-Exactive HF was operated in data-dependent mode with survey scans acquired in the Orbitrap mass analyzer over the range of 380–1800 m/z with a mass resolution of 120,000. MS/MS spectra were performed selecting the top 15 most abundant +2, +3, or +4 ions and precursor isolation window of 1.2 m/z. Selected ions were fragmented by Higher-energy Collisional Dissociation (HCD) with normalized collision energies of 28, and the mass spectra acquired in the Orbitrap mass analyzer with a monitored first mass of 100 m/z, mass resolution of 15,000, AGC target set to $1 \times 10^5$, and max injection time set to 100 ms. A dynamic exclusion window was set for 30 s.

## Phosphoproteomic data analysis

The peptide identification and quantification pipeline relied on the Trans Proteomic Pipeline (v. 5.2.0) and the Comet search engine (v. 2019.01 revision 5) (*Eng et al., 2013*). The Mouse UniProt proteome database (22,297,478 entries) was downloaded on 22-10-2018. Search parameters included tryptic requirement, 10 ppm for the precursor match tolerance, dynamic mass modification of 79.966331 Da for phosphorylation of serine, threonine, and tyrosine and static mass modification of 57.021465 Da for alkylated cysteine residues. A static N-terminal TMT isobaric tagging modification of 229.162932 was also applied, alongside a dynamic modification of 15.9949 for oxidation of methionine. TMT-labeling correction parameters were entered into a Libra conditions file according to the information provided by the manufacturer. All additional Comet parameters were left at their default values. PeptideProphet was used to validate peptide identifications, and Libra was used to quantify TMT reporter ion intensities from which quantitative ratios were calculated. The phosphoproteomic data generated in this study were deposited to the MassIVE database (http://massive.ucsd.edu) and received the ID: MSV000086764, doi:10.25345/C57N54, and ProteomeXchange ID: PXD023803. Following processing by Comet, PeptideProphet, Libra, and PTMProphet, phosphoproteomic datasets were exported as .xls files for further processing via R scripts. A first script calculated fold changes between TMT reporter ion channels using the median of Libra-provided values for each condition. This script eliminated any peptides that featured missing reporter ion intensity values for channels in both conditions for a given experiment. A second script normalized the fold changes and procedurally clustered phosphorylation sites if PTMProphet localized them to adjacent phosphorylatable residues. These clustered sites' localization probabilities were summed, and if this cumulative score remained above 0.85, the phosphorylation sites were designated a valid cluster and remained in the dataset for further analysis. After quantification, normalization, and clustering of each experimental dataset, AZ20-treated and *Rad1* cKO experiments were compared with one another such that sites seen in both sets of experiments could have their average fold changes plotted against each other on Cartesian coordinates. Quadrants were designated as Q1–Q4 delimited by an interval of correlation correspondent to fivefold of the log2 scale. Any sites whose average ratios fell within a circle of radius of 0.7-fold change centered on the origin were designated as 'Center' sites and were considered unregulated. Phosphorylation sites placed into quadrants were subjected to a second filtering step to account for any inconsistent ratios between replicates of each experiment. This filter removed any

site that featured at least one inverted (opposite of expected quadrant sign) experimental ratio with a magnitude greater than or equal to 0.25-fold change.

## Meiotic chromosome spread immunofluorescence

Meiotic spreads were prepared as previously described (*Holloway et al., 2014*). Briefly, decapsulated testes tubules were incubated in a hypotonic extraction buffer (30 mM Tris pH 7.2, 50 mM sucrose, 17 mM citrate, 5 mM EDTA, 0.5 mM DTT, 0.1 mM PMSF) for 1 hr. 1 mm sections of tubule were dissected in 100 mM sucrose solution and then added to slides coated in 1% paraformaldehyde/0.15% Triton X and allowed to spread for 2.5 hr in a humidification chamber. Slides were then dried for 30 min and washed in 0.4% photoflo (Kodak)/PBS solution for 5 min. Slides were immediately processed for immunofluorescence or frozen at –80°C. For staining, slides were washed in a solution of 0.4% photoflo/PBS for 10 min, 0.1% Triton X/PBS for 10 min and blocked in 10% antibody dilution buffer (3% BSA, 10% goat serum, 0.0125% Triton X)/PBS for 10 min. Primary antibodies were diluted in antibody dilution buffer at the indicated dilution and incubated with a strip of parafilm to spread the antibody solution in a humidification chamber at 4°C overnight. After primary antibody incubation, the slides are washed with 0.4% photoflo/PBS for 10 min, 0.1% Triton X for 10 min, and blocked with 10% antibody dilution buffer. Secondary antibodies were diluted as indicated and incubated on slides with a parafilm strip at 37°C for 1 hr. Slides were then washed in 0.4% photoflo/PBS for 10 min twice followed by 0.4% photoflo/H$_2$O for 10 min twice and allowed to dry before mounting with DAPI/antifade. Slides were imaged on a Leica DMi8 Microscope with a Leica DFC9000 GTC camera using the LAS X (Leica Application Suite X) software. For every condition, a minimum of 50 images from three independent mice were acquired. To quantify florescence intensity, the LAS X software quantification tool was used. Briefly, a region of interest (ROI) line was drawn over the sex body and mean intensity of the underlying pixels was recorded. Additionally, two ROI lines of equal length were placed over two autosome cores and the mean pixel intensity was also recorded to serve as an internal control for background florescence. The sex body ROI intensity was then normalized to the average of the two autosomal ROI intensities for each individual cell. For the autosomal signal CDK2, a ROI was drawn along each autosomal core as identified by SYCP3 staining. The signal intensity for each autosome was then averaged per cell. For the *Rad1* cKO mice, where a true sex body does not form, a line was drawn over chromosomes that were best morphologically identified as the sex chromosomes by SYCP3 staining.

## Hematoxylin and eosin staining

Adult testes were dissected and incubated in Bouin's fixative for 7 hr, washed 4× in 70% ethanol and embed in paraffin. 5 mM sections were mounted on slides and rehydrated in SafeCear Xylene Substitute followed by decreasing amounts of ethanol. Slides were then stained with hematoxylin followed by eosin and gradually dehydrated by incubation in increasing concentrations of ethanol before mounting using permount mounting medium.

## Antibody list

| Reagent – secondary antibody | Identifier | Dilution |
|---|---|---|
| Goat anti-mouse Alexa Fluor 488 | 62-6511 | 1:2000 |
| Goat anti-mouse Alexa Flour 555 | A-10521 | 1:2000 |
| Goat anti-rabbit Alexa Fluor 488 | 65-6111 | 1:2000 |
| Goat anti-rabbit Alexa Fluor 555 | A-10520 | 1:2000 |

| Reagent – primary antibody | Source | Identifier | Dilution |
|---|---|---|---|
| yH2AX | Millipore | 05-636 | 1:10,000 |
| pMDC1 (phosphoT4) | Abcam | Ab35967 | 1:500 |
| SETX | Abcam | Ab220827 | 1:100 |

*Continued on next page*

*Continued*

| Reagent – primary antibody | Source | Identifier | Dilution |
|---|---|---|---|
| RANBP3 | Bethyl | IHC-00295 | 1:100 |
| SYCP3 (mouse) | Custom (*Kolas et al., 2005*) | N/A | 1:2000 |
| SYCP3 (rabbit) | Abcam | Ab15093 | 1:2000 |
| CDK2 (M2) | Santa Cruz | SC-163 | 1:100 |

## Acknowledgements

We thank all members of the Smolka, Weiss, and Cohen Labs for valuable discussions. We thank Robert Gingras and Mason Muir for assistance with imaging and image quantification, Fenghua Hu and Tony Bretscher for use of the microscopes, and Shannon Marshall for assistance with phosphopeptide enrichment. Figures 1B and Figure 2—figure supplement 4A were created using BioRender.com. This work was supported by a grant from the National Institute of Health (R01-HD095296) to MBS and RW, grants from the Spanish Ministry of Innovation Science and Universities/ EU-ERDF (PID2019-109222RB-I00) and Agencia Canaria de Investigación, Innovación y Sociedad de la Información/ EU-ERDF (ProID2020010109) to RF, and a grant from the National Institute of Health (5R01HD097987) to PEC.

## Additional information

### Funding

| Funder | Grant reference number | Author |
|---|---|---|
| Eunice Kennedy Shriver National Institute of Child Health and Human Development | R01-HD095296 | Robert S Weiss Marcus B Smolka |
| Spanish Ministry of Innovation Science and Universities | PID2019-109222RB-I00 | Raimundo Freire |
| Agencia Canaria de Investigación | ProID2020010109 | Raimundo Freire |
| Eunice Kennedy Shriver National Institute of Child Health and Human Development | 5R01HD097987 | Paula E Cohen |

The funders had no role in study design, data collection and interpretation, or the decision to submit the work for publication.

### Author contributions

Jennie R Sims, Conceptualization, Data curation, Formal analysis, Investigation, Methodology, Resources, Supervision, Validation, Visualization, Writing – original draft, Writing – review and editing; Vitor M Faça, Data curation, Formal analysis, Methodology, Supervision, Writing – review and editing; Catalina Pereira, Generation of Rad1-CKO mouse, Resources; Carolline Ascenção, Jumana Badar, Data curation, Formal analysis; William Comstock, Data curation, Formal analysis, Methodology; Gerardo A Arroyo-Martinez, Data curation, Resources, Software; Raimundo Freire, Resources; Paula E Cohen, Data curation, Funding acquisition, Resources, Supervision, Writing – review and editing; Robert S Weiss, Funding acquisition, Project administration, Resources, Supervision, Writing – review and editing; Marcus B Smolka, Conceptualization, Data curation, Formal analysis, Funding acquisition, Project administration, Resources, Supervision, Visualization, Writing – original draft, Writing – review and editing

## Author ORCIDs

Jennie R Sims (iD) http://orcid.org/0000-0002-1882-7511
Catalina Pereira (iD) http://orcid.org/0000-0003-3144-0909
Raimundo Freire (iD) http://orcid.org/0000-0003-4473-8894
Paula E Cohen (iD) http://orcid.org/0000-0002-2050-6979
Robert S Weiss (iD) http://orcid.org/0000-0003-3327-1379
Marcus B Smolka (iD) http://orcid.org/0000-0001-9952-2885

## Ethics

All mice used for this study were handled following federal and institutional guidelines under a protocol (2004-0034) approved by the Institutional Animal Care and Use Committee (IACUC) at Cornell University.

## Decision letter and Author response

Decision letter https://doi.org/10.7554/eLife.68648.sa1
Author response https://doi.org/10.7554/eLife.68648.sa2

---

## Additional files

### Supplementary files

• Supplementary file 1. Complete phosphoproteomic dataset of the 4 hr ATRi vs. *Rad1* cKO experiments. Complete list of phosphopeptides identified in each biological replicate for mice treated with ATRi for 4 hr and the *Rad1* cKO conditions. Column header definitions can be found in sheet 1. See Materials and methods for details on data processing and filtering. Phosphosites removed by filtering can be found in the last tab.

• Supplementary file 2. Gene ontology STRING analysis output. Gene ontology information from STRING-db for subsets of data from the 4 hr or 2.5–3-day ATRi treatment combined to *Rad1* cKO. Each quadrant or type of filter applied is indicated on the tab name.

• Supplementary file 3. Complete phosphoproteomic database of the 2.5–3-day ATRi vs. *Rad1* cKO experiments. Complete list of phosphopeptides identified in each biological replicate for mice treated with ATRi 2.5 and 3 days hours and the *Rad1* cKO conditions. *Rad1* cKO phosphopeptides are the same presented in *Supplementary file 1*. Column header definitions can be found in sheet 1. See Materials and methods for details on data processing and filtering. Phosphosites removed by filtering can be found in the last tab.

• Supplementary file 4. Phosphoproteomic database comparison of Q2 4 hr ATRi and 2.5–3-day ATRi. Comparison of the Q2 phosphopeptides found in the 4 hr and/or 2.5–3-day ATRi treatment conditions. Data is separated into Q2 sites exclusively found in the 4 hr ATRi treatment, 2.5–3 hr ATRi treatment or sites common between both datasets, separated into different sheets.

• Supplementary file 5. Gene ontology chord analysis output. Gene ontology output from chord diagrams. See Materials and methods for more details.

• Transparent reporting form

### Data availability

The phosphoproteomic data generated in this study were deposited to the Massive database (http://massive.ucsd.edu) and received the ID: MSV000086764, https://doi.org/10.25345/C57N54, and ProteomeXchange ID: PXD023803.

The following dataset was generated:

| Author(s) | Year | Dataset title | Dataset URL | Database and Identifier |
|---|---|---|---|---|
| Vitor F, Smolka M | 2021 | Phosphoproteomics of ATR Signaling in Prophase I of Mouse Meiosis | https://massive.ucsd.edu/ProteoSAFe/dataset.jsp?accession=MSV000086764 | The phosphoproteomic data generated in this study were deposited to the MassIVE database and received the ID: MSV000086764, doi:10.25345/C57N54, and ProteomeXchange ID: PXD023803, MSV000086764 |

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
