## [Editor Report]

This paper nicely provides a valuable list of phosphorylation targets by the DNA damage checkpoint kinase, ATR, during mouse spermatogenesis. Importantly, by using the dataset, the authors now showed that ATR controls the localization of CDK2 on meiotic chromosomes, which is critical for meiotic crossover formation.

---

## [Decision Letter]

**Decision letter after peer review:**

Thank you for submitting your article "Phosphoproteomics of ATR Signaling in Prophase I of Mouse Meiosis" for consideration by *eLife*. Your article has been reviewed by 3 peer reviewers, including Akira Shinohara as Reviewing Editor and Reviewer #1, and the evaluation has been overseen by Jessica Tyler as the Senior Editor.

The Reviewing Editor and other two reviewers all acknowledge the importance of this work and recommend possible publication in *eLife* as a "Tools and Resource" paper. However, we all agree that you need a major revision of your paper, particularly in respect to the relationship with the co-submitted paper with Dr. Weiss.

Since using the same data sets in co-submitted papers is unusual, we ask you to integrate most of the ATR-dependent phospho-proteomic data described in Figure 6 (and possibly Figure 7 since there is little insight on the role of ATR-dependent phosphorylation of Smc3) from the accompanying paper by Weiss into your paper. Importantly, to strengthen your conclusion in the paper, we recommend you to re-analyze your data sets. Below are the list of experiments and additional analysis that we would like you to add in the revised version. We would like revisions to be returned within two months. If the revisions take a lot longer, you may want to resubmit as a new manuscript, and we will make sure that it goes to the same editors and reviewers.

Essential points (experiments):

1. One major issue is on the different cellular compositions between controls and Rad1-conditonal knockout (cKO) testes. The authors likely detected developmental changes in phosphorylation sites in spermatogenesis, and the authors are aware of this caveat (Line 160-165). To circumvent this issue, the authors may perform phosphoproteomics of first-wave Rad1-cKO testes (~ postnatal day 14 or 15), where the cellular composition of Rad1-cKO testes was not grossly changed from that of controls. They should also show the testicular section data and confirm the cellular composition of the testis they analyze. You may add these data set as Supplement; Duplicate would be preferred, but not essential.

2. There are two different experiments for ATRi-treated mice (2 pairs after 2.5-3 days of treatment and 2 pairs 4 hours after a single dose). However, these results are not distinguished in the analysis, and there is no evaluation of testicular morphology in both cases after the ATRi treatment. The authors should analyze these data separately as the experiments were distinct. Also, they should show the testicular morphology after the ATRi treatment and discussed the difference a little bit more.

3. It is very interesting hypothesis raised by the authors that ATR may control the activity of other kinases and phosphatases which will be responsible for most of the phosphorylation site changes observed in this study. To follow up on this, the author may check the localization of CDK1/2, MAK, NEK1, PKMYT1, PPM1G, and/or PP1R7 in control, ATRi-treated, and RAD1cKO spermatocytes? For some of these proteins, commercial antibodies are available, even for their phosphorylated forms.

Other points (text editing etc);

1. Consider changing the title of the manuscript since you are studying the ATR signaling in mouse testis, not only meiotic cells.

2. The abstract was overstated. The study unbiasedly identified any phosphorylation sites. The abstract is misleading because "12,000 phosphorylation sites" are all phosphorylation sites detected in this study. This is not surprising, as previous studies have already detected many more phosphorylation sites (17,829 sites in Qi et al., MCP 2014, and 13,835 sites in Li et al., in Proteomics 2019). The sentence "Here we defined ATR signaling during prophase I in mice" may be an overstatement.

3. The introduction section contains several errors:

4. Ref #10 did not examine meiosis.

5. Line 49: CHK1 is not required for MSCI, and CHK1 and BRCA1 do not spread from the axes to chromatin loops.

6. Some references for ATR's function in meiosis are missing. Please cite Fedoriw, et al., Development 2015 and Menolfi, et al., Nat Commun 2018.

7. A study showed that ETAA1 unlikely functions in meiosis (Ellnati, et al., PNAS 2018). Discussion about TEAA1 can be adjusted.

8. There are numerous errors in figure numbers, and these results are difficult to evaluate.

9. Table S1 was referenced in many places, but it is impossible to interpret the described data. Table S1 is just a list of ~12,000 sites. Appropriate data should be shown for each description.

10. Figure 2G is not described in the text.

11. The entire section following "Connectivity analysis defines ATR-regulated sub-networks" does not make sense. Figure 3A is not STRING analysis but is described so in the text. All figure numbers for sup data are misaligned. Line 213: there is no data about "cell cycle" in the figure.

12. The data pertaining to ATR-dependent SETX and RANBP3 are interesting, but the authors cannot conclude if this is phosphor-dependent. Similarly, they cannot conclude that "ATR-dependent phosphorylation promotes the recruitment or retention of Senataxin at the sex body" (Line 298).

13. Table S1 Please provide the protein name also apart from protein ID.

14. Revise supplemental figure citation because I think most of them are incorrect.

15. Figure 3E. I would like the authors to either refer to the Phospho-S/T-D/E sites in the main text or simplified the figure by pooling these with the other phosphorylation sites.

16. I have missed in the results, or the discussion, the mention of one of the best known ATR targets (CHK1). Was this protein found in the analysis (I could not find it in Table S1)? If not, how do the authors explain this result?

17. Although the authors used spermatocytes to identify the ATR-dependent targets, they have not mentioned the phosphorylation of meiosis (spermatocytes)-specific proteins. Moreover, the authors described mainly targets which have been identified in mitotic cells such as MDC1, TOPBP1, and CTIP etc. It would be interesting to compare phospho-proteome of spermatocytes with that of mitotic cells with exogeneous DNA double-strand breaks (DSBs).

18. Given that ATR belongs to the PI3 kinase family, in some senses, ATR is, in some cases, redundant to ATM kinase (and DNA-PK), it would be very nice to talk about the specificity of ATRi, AZ20. This is very important since there is only weak enrichment of ATM/ATR consensus sequence, S/TQ in the data set.

19. It would be nice to show what kind of meiotic defects are induced after acute and chronic treatment of ATRi such as apoptosis or impaired progression of meiosis.

---

## [Author Response]

Essential points (experiments):1. One major issue is on the different cellular compositions between controls and Rad1-conditonal knockout (cKO) testes. The authors likely detected developmental changes in phosphorylation sites in spermatogenesis, and the authors are aware of this caveat (Line 160-165). To circumvent this issue, the authors may perform phosphoproteomics of first-wave Rad1-cKO testes (~ postnatal day 14 or 15), where the cellular composition of Rad1-cKO testes was not grossly changed from that of controls. They should also show the testicular section data and confirm the cellular composition of the testis they analyze. You may add these data set as Supplement; Duplicate would be preferred, but not essential.

We thank the reviewers for this suggestion. We agree that the differences in cellular composition is a convoluting factor when comparing the *Rad1* cKO and WT testes. We believe that our dual approach of only considering sites that are depleted in *both* the *Rad1* cKO *as well as the 4 hour ATRi treated mice* (which do not have changes in cellular composition) (see point 2) addresses this concern. We do acknowledge that it would be ideal to perform the phosphoproteomic experiments in *Rad1* cKO 14 day-old mice, before significant cell loss has occurred. Unfortunately, the amount of tissue required to perform phosphoproteomic analysis on first-wave *Rad1* cKO testes would be prohibitive and we were not able to perform this experiment. However, with the limited number of *Rad1* cKO mice available to us, we were able to confirm that SETX and RANBP3 localization in 14-day old *Rad1* cKO testes was not different compared to 8-12 week *Rad1* cKO animals from our initial analysis, supporting the validity of our approach (please see Figure 4—figure supplement 1D and Figure 4—figure supplement 2D).

2. There are two different experiments for ATRi-treated mice (2 pairs after 2.5-3 days of treatment and 2 pairs 4 hours after a single dose). However, these results are not distinguished in the analysis, and there is no evaluation of testicular morphology in both cases after the ATRi treatment. The authors should analyze these data separately as the experiments were distinct. Also, they should show the testicular morphology after the ATRi treatment and discussed the difference a little bit more.

We thank the reviewers for this important suggestion. We have now analyzed the 2.5-3 day ATRi treatment separately from the 4 hr ATRi datasets. Importantly, we repeated the 4 hr treatment condition (now five replicates) to enhance our confidence in this dataset. We have included the 2.5-3 day ATRi dataset analysis in separate supplemental figures and tables, and have used it for the analysis of the S/T-P-X-K motif and to compare the changes in this motif over the 4hr and 2.5-3 day inhibition period. We have confirmed that tubule morphology and cellular composition is not altered in the 4 hr ATRi treatment condition and have added this data to the supplement.

3. It is very interesting hypothesis raised by the authors that ATR may control the activity of other kinases and phosphatases which will be responsible for most of the phosphorylation site changes observed in this study. To follow up on this, the author may check the localization of CDK1/2, MAK, NEK1, PKMYT1, PPM1G, and/or PP1R7 in control, ATRi-treated, and RAD1cKO spermatocytes? For some of these proteins, commercial antibodies are available, even for their phosphorylated forms.

We thank the authors for the positive comment on the finding that ATR is likely regulating other downstream kinases and phosphatases. We thank the suggestion of monitoring potential localization changes of some of the kinases/phosphatases upon ATR inhibition. We performed a series of IF experiments to assess the localization of several kinases/phosphatases for which antibodies were available. Despite multiple attempts, we could not detect specific signal in meiotic spreads for several of the proteins in the vehicle or 4 hour ATRi treatment condition. Excitingly, we were able to observe specific signal for CDK2 and found that its localization in meiotic spreads was significantly reduced after 4 hours of ATRi treatment (New Figures 5C-D and Figure 5—figure supplement 2). We believe that this is an important finding that highlights the potential for ATR activity to regulate CDK localization and allow CDK-dependent phosphorylation of several targets, consistent with our finding that ATRi treatment results in the loss of several phosphorylation events at the S/T-P-X-K motif (Figure 5A-B). We have included this new data to a remodeled Figure 5 and updated the discussion accordingly.

Other points (text editing etc);1. Consider changing the title of the manuscript since you are studying the ATR signaling in mouse testis, not only meiotic cells.

We agree with this reviewer comment since our analysis was done on whole testes and not purified meiotic cell populations. We have changed the title of the manuscript to “Phosphoproteomics of ATR signaling in Mouse Testes”.

2. The abstract was overstated. The study unbiasedly identified any phosphorylation sites. The abstract is misleading because "12,000 phosphorylation sites" are all phosphorylation sites detected in this study. This is not surprising, as previous studies have already detected many more phosphorylation sites (17,829 sites in Qi et al., MCP 2014, and 13,835 sites in Li et al., in Proteomics 2019). The sentence "Here we defined ATR signaling during prophase I in mice" may be an overstatement.

We thank the reviewer for this feedback and have revised the abstract to more accurately describe our approach and results. We now emphasize that while we identified more than 14,000 phosphorylation sites, with 401 of these sites being dependent on ATR and RAD1. Furthermore, we have cited the publications mentioned by the reviewer.

3. The introduction section contains several errors:

We have carefully revised all the text in the manuscript, including introduction.

4. Ref #10 did not examine meiosis.

Thank you for this correction, we have removed reference 10.

5. Line 49: CHK1 is not required for MSCI, and CHK1 and BRCA1 do not spread from the axes to chromatin loops.

We thank the reviewer for this correction and have updated the text to correct this error.

6. Some references for ATR's function in meiosis are missing. Please cite Fedoriw, et al., Development 2015 and Menolfi, et al., Nat Commun 2018.

We thank the reviewer for this correction and have added these references.

7. A study showed that ETAA1 unlikely functions in meiosis (Ellnati, et al., PNAS 2018). Discussion about TEAA1 can be adjusted.

We agree that ETAA1 is unlikely to function in meiosis, although we cannot exclude that as a possibility in our system. We have updated the introduction and discussion to reflect this.

8. There are numerous errors in figure numbers, and these results are difficult to evaluate.

We apologize for the errors. We thank the reviewers for pointing this and have corrected figure numbers.

9. Table S1 was referenced in many places, but it is impossible to interpret the described data. Table S1 is just a list of ~12,000 sites. Appropriate data should be shown for each description.

We have added more specific referencing to the data. We have also added more comprehensive table descriptors to Supplementary Files 1 and 3, which contain extensive relevant data for the descriptors in the text, including sequence motif for each site, designated quadrant, etc.

10. Figure 2G is not described in the text.

We thank the authors for this correction and have updated the text to describe this data.

11. The entire section following "Connectivity analysis defines ATR-regulated sub-networks" does not make sense. Figure 3A is not STRING analysis but is described so in the text. All figure numbers for sup data are misaligned. Line 213: there is no data about "cell cycle" in the figure.

To improve clarity, we have completely remodeled Figure 3 and related text. Parts of it were moved to supplemental data.

12. The data pertaining to ATR-dependent SETX and RANBP3 are interesting, but the authors cannot conclude if this is phosphor-dependent. Similarly, they cannot conclude that "ATR-dependent phosphorylation promotes the recruitment or retention of Senataxin at the sex body" (Line 298).

We believe that our results strongly suggest that the localization of SETX and RANBP3 is dependent on ATR signaling since the localization is lost in both the ATRi treated mice and *Rad1* cKO mice. Moreover, both SETX and RANBP3 have identified S/T-Q sites that are also lost in ATRi treated mice and *Rad1* cKO mice. For the sake of extra caution, we have modified our conclusions to use the words “strongly suggest”.

13. Table S1 Please provide the protein name also apart from protein ID.

We agree and have added this to Supplementary Files 1 and 3.

14. Revise supplemental figure citation because I think most of them are incorrect.

Thanks, we have corrected these errors.

15. Figure 3E. I would like the authors to either refer to the Phospho-S/T-D/E sites in the main text or simplified the figure by pooling these with the other phosphorylation sites.

Done. Of note, previous Figure 3E is now Figure 5F. Also, we have completely remodeled previous Figure 3, which is now Figure 5 focused mostly on S/T-X-P-K sites and the new finding that ATR inhibition alters localization of CDK2.

16. I have missed in the results, or the discussion, the mention of one of the best known ATR targets (CHK1). Was this protein found in the analysis (I could not find it in Table S1)? If not, how do the authors explain this result?

We appreciate the reviewer bringing this to our attention. We did not detect CHK1 in our analysis, which we believe reflects technical challenges in achieving complete coverage of the phosphoproteome and the potential low abundance of CHK1 in spermatocytes. We have updated the discussion to address this concern and have also reiterated that we did identify other known ATR targets such as MDC1 and yH2AX.

17. Although the authors used spermatocytes to identify the ATR-dependent targets, they have not mentioned the phosphorylation of meiosis (spermatocytes)-specific proteins. Moreover, the authors described mainly targets which have been identified in mitotic cells such as MDC1, TOPBP1, and CTIP etc. It would be interesting to compare phospho-proteome of spermatocytes with that of mitotic cells with exogeneous DNA double-strand breaks (DSBs).

We appreciate this suggestion but feel this is beyond the scope of this study. We believe future work will address this point.

18. Given that ATR belongs to the PI3 kinase family, in some senses, ATR is, in some cases, redundant to ATM kinase (and DNA-PK), it would be very nice to talk about the specificity of ATRi, AZ20. This is very important since there is only weak enrichment of ATM/ATR consensus sequence, S/TQ in the data set.19. It would be nice to show what kind of meiotic defects are induced after acute and chronic treatment of ATRi such as apoptosis or impaired progression of meiosis.

We thank the reviewer for bringing this point and have added additional citations to previous work where the consequences of ATRi treatment for 3-7 days was discussed, showing loss of spermatocyte populations arising from DNA repair, chromosome synapsis and sex body formation defects. We also discuss the point about specificity of AZ20, which is part of a new generation of highly specific inhibitors of ATR, therefore not expected to interfere with a range of S/T-Q sites targeted by ATM or DNA-PKcs, as pointed out by the reviewer.